# How Refugees Experience the Australian Workplace: A Comparative Mixed Methods Study

**DOI:** 10.3390/ijerph18084023

**Published:** 2021-04-12

**Authors:** Patricia Cain, Alison Daly, Alison Reid

**Affiliations:** School of Population Health, Curtin University, Kent Street, Bentley, WA 6102, Australia; p.cain@edu.edu.au (P.C.); alison.daly@curtin.edu.au (A.D.)

**Keywords:** workplace exposure, migrants, refugees, psychosocial hazards, mixed-methods

## Abstract

There is a growing body of evidence indicating poorer working conditions for migrant workers, particularly refugees, compared with native-born workers. Our objectives were to compare exposure to workplace psychosocial stressors in working refugees with other migrant groups and Australian-born workers of Caucasian ancestry and to describe the working experience of refugees. Cross-sectional surveys collected information on the workplace stressors of job complexity, control, security, bullying, and racial discrimination from six migrant groups (n = 1062) and Caucasian Australians (n = 1051); semi-structured face-to-face interviews were used with currently employed refugees (n = 30). Content analysis examined the qualitative data. Compared to all other groups, working refugees were more likely to report experiencing racial discrimination in the workplace and to report exposure to more than three hazards. Content analysis indicated that working refugees are working below their capacity, in terms of hours and qualifications, and in jobs that were low status and lacked security. Despite challenging work conditions, participants reported adequate health and safety training and feeling a sense of pride in their work. These findings highlight the need for better support for refugees in negotiating the workplace once they find employment and the importance of employers providing an inclusive and equitable workplace.

## 1. Introduction

Refugees and asylum seekers make up an increasing component (29%) of the global migrant population of 272 million [1]. The United Nations High Commissioner for Refugees (UNHCR) estimates that there are 79.5 million forcibly displaced persons, consisting of 45.7 million internally displaced people, 26 million refugees (those who have had their claim for refugee status recognised under the 1951 Convention), and 4.2 million asylum seekers (those who have not yet had their claim for refugee status recognised). Sixty-eight percent of refugees currently come from five countries: Syria, Venezuela, Afghanistan, South Sudan, and Myanmar [2]. Australia currently settles over 12,000 refugees annually as part of its humanitarian program, and it has done so in varying numbers since the first 170,000 people were accepted as part of the displaced persons program established at the end of World War II [3]. In the 2018–2019 period, Iraq, the Democratic Republic of Congo, Myanmar, Syria, and Afghanistan were the top five sending countries [4]. Most people arrived in Australian from South Sudan during the period 2001–2006 [5].

Internationally, including Australia, there is a growing body of evidence showing poorer working conditions among immigrant/migrant than native-born workers and a consequent increase in work-related injuries and occupational diseases [6,7,8]. Migrant workers are more likely to work in jobs that are characterised by high complexity and low control [9,10] and in jobs that are insecure and precarious [11]. Other work from Australia reports disparities in exposure to carcinogens among migrant and native-born workers [12,13]. Furthermore, a recent survey in Australia reported that one-third of temporary migrant workers in Australia, principally backpackers and international students, incur wage theft, earning about half of the legal minimum wage [14].

Specifically, this research does not tend to examine the working conditions of refugee workers (e.g., refugees who are working in their host country), although there is a larger body of work concerning the difficulties refugee workers have obtaining work [15,16,17], but that is not the focus of this study. There is a range of reasons why refugees might be more likely to incur poorer working conditions compared with other groups of migrants (e.g., skilled migrants, temporary migrants, or international students) or native-born workers. These include segregation into niche occupations and industries [18,19] and non-recognition of prior educational qualifications [20,21], lack of mainstream social networks to assist in job searching, poorer English language proficiency, as well as the sending home of remittances necessitates risk taking and working in survival jobs [6].

The aim of this mixed methods study was to compare the working conditions of refugee workers in Australia with those of workers of Vietnamese, Chinese, and Arabic speaking ancestry, workers born in India, the Philippines, and New Zealand [22,23], and Australian-born workers of Caucasian ancestry [12,13,24,25]. The second objective was to learn more about how refugee workers experience the workplace. To do this, participants spoke freely about the conditions they experience and how they respond to those conditions.

## 2. Materials and Methods

### 2.1. Participant Recruitment

Men and women born in Afghanistan, Iraq, and South Sudan, who had arrived in Australia as a refugee or asylum seeker, were currently working, and able to be interviewed in English were invited to participate. Sampling was purposive, and recruitment started by approaching 57 community organisations with links to refugee and migrant communities. Organisations were contacted through email, social media, and in person. In addition, a community leader with access to the populations of interest was employed to assist in recruitment. Participant recruitment ceased when no new information was obtained by completing more interviews [26].

The nature of the study and the interview process, including the recording of interviews, was explained to participants so that they were able to provide informed consent to take part in the project. Semi-structured face-to-face interviews were conducted at a time and place convenient to participants, with most interviews conducted in participant’s homes. Interviews ranged in length from 40 min to one hour 20 min. Participants were assured of anonymity and confidentiality and received a $50 shopping voucher as compensation for their time given to the project. Participants were also provided a list of contact numbers for organisations and resources should they have any work-related or migration issues that they wished to pursue. Interviews were conducted, recorded, and transcribed verbatim by the first author. Ethics approval for this study was obtained from the Human Research Ethics Committee of Curtin University.

The quantitative comparison involved a secondary analysis of existing data. The information was taken from three previously conducted cross-sectional telephone surveys investigating occupational health and safety among migrant workers in Australia. Survey one (2015) interviewed migrant workers from Chinese, Vietnamese, and Arabic speaking backgrounds (n = 595). Survey two (2017–2018) interviewed migrant workers born in New Zealand, India, and the Philippines (n = 1630), and survey three (2017) interviewed Australian-born workers (Caucasian only included in this current study; n = 1051). These groups of workers were chosen as suitable comparisons to the working refugee group for several reasons. (1) India, China, Vietnam, the Philippines, and New Zealand are in the top 10 group of countries Australia receives migrants from. (2) Earlier focus group discussions we had undertaken highlighted that workers of Arabic-speaking ancestry were particularly disadvantaged in the workplace, in terms of exposure to hazards, and this was confirmed in our quantitative survey [12,13]. (3) Similarly, earlier work of ours had shown that workers born in New Zealand had higher hospitalisations for work-related injuries and higher work-related fatalities than Australian-born workers or workers born in any other country [23]. New Zealanders enter Australia on a different visa type than the skilled workers coming from China, Vietnam, the Philippines, and India. (4) We included an Australian-born group of Caucasian ancestry because in our earlier focus group discussions and in-depth interviews, second-generation migrants, of non-Caucasian ancestry, reported barriers in the workplace that they perceived did not occur among those with Caucasian ancestry [6].

### 2.2. Data Collection

A full description of the methods for these studies has been published elsewhere, but briefly, Survey One (conducted in 2015, n = 535) used quota sampling from a mix of telephone lists and lists from sample brokers to recruit participants in the capital cities of Perth, Sydney, and Melbourne; Survey Two (conducted in 2016–2017, n = 1062) used random digit dialing of telephone lists to recruit participants across Australia; and Survey Three (conducted in 2017, n = 1630) used a combination of probability and non-probability sampling including random sampling from telephone lists supplemented with census data to identify target group high-density suburbs across Australia. In Survey Three, the information obtained from these earlier quantitative surveys was used to inform the development of the qualitative survey among refugee workers. Initially, we thought that refugee workers could be recruited using respondent-driven sampling across Australia, but we were unable to recruit sufficient numbers of seeds necessary in each state for this process to work. Furthermore, our key informants and stakeholders advised that this method was not appropriate for this population. So, instead, we limited our recruitment to Western Australia and worked with key informants here to recruit participants. Some of the questions used in the quantitative surveys were adapted for use qualitatively, and these formed the basis of comparison with the quantitative data.

In order to compare refugee worker responses with earlier results from the three cross-sectional surveys, we asked eleven questions used in those surveys, which were taken from the Household, Income and Labour Dynamics in Australia survey (HILDA) [27] (Table 1). Responses were rated on a seven-point Likert scale ranging from strongly agree to strongly disagree. Full details on how these measures were combined to form estimates of job complexity, control, security and job quality have been reported previously [24]. A higher mean score for complexity indicates high job complexity, higher mean scores for control, security, and job quality indicate more job control, good job security. The job complexity score was reversed and combined with the control, security, and unfair pay scores to identify an overall job quality mean, with a high score indicating high job quality [28,29,30].

Refugee participants were asked about bullying and discrimination in the workplace. (Responses for perceived bullying and discrimination were coded as yes/no. These responses were compared with questions collected in the three cross-sectional surveys in which respondents were asked if they had ever been bullied within the last six months in the workplace and been discriminated against in the workplace due to race or ethnicity in the last year.

As well as the structured questions outlined above, open-ended questions were asked to capture more in-depth information about participants’ workplace experiences (Table 1). Quantitative and qualitative data were collected simultaneously. These questions were based on aspects of work that may impact on well-being or potentially mediate response to adverse events and situations. Some questions were follow-up enquiries based on responses to survey items; e.g., for participants who had experienced bullying or discrimination at work, follow-up questions on the frequency and duration of these experiences were asked. In addition, four questions asked whether (and how) participants responded to unfair treatment at work, and four questions asked about relationships with supervisors and co-workers. Other questions were derived from previous findings on the working conditions and challenges for migrant workers [6] and refugee workers [17,31,32]. To indicate underemployment, that is whether participants were working in jobs that were below their capacity in terms of hours worked, we asked if participants would prefer to work more hours. To indicate over-qualification in relation to education, we asked if participants had skills and qualifications that they were not using in their current job. To understand more about the rewarding aspects of work, we asked two questions on whether participants felt valued and respected at work, and whether they felt a sense of pride for the work they did [33]. Three questions asked about health and safety at work and four asked about financial pressures, wage theft, and remittances. Other questions asked about finding work and doing volunteer work [34]. Demographic questions (e.g., age, sex, occupation, educational level) were asked last.

### 2.3. Analysis

The three cross-sectional surveys were weighted using a technique that weights the sample to the same proportions found in each migrant population (Iterative Proportional Fitting) [35]. We weighted each migrant group for sex, age, education, and area of residence to produce population estimates of workplace psychosocial stressors. Due to sample size differences, percentages and means with 95% confidence intervals were produced for comparison with the current study results. The confidence intervals were used to indicate statistically significant differences. The mixed method approach used in this study was that of triangulation design: convergence model, which does not require the integration of the different datasets for analysis [36]. We used semi-structured interviews to collect qualitative data and then compared the results from the qualitative interviews with the results we already had from previous cross-sectional studies [37].

For the qualitative analysis, responses to the open-ended interview questions were analysed using content analysis. As a method, content analysis allows non-numeric data to be accounted for quantitatively [38]. Our approach to qualitative analysis was deductive; the topics of the interview questions formed the basis of coding. A review of the topic guide identified 33 codes. Then, transcripts were examined, and explicit content was allocated to mutually exclusive codes [39]. Initial codes were reviewed and refined into 21 content areas and grouped into six broad categories. Coding was conducted by the first author, with the third author reviewing coding and content classification.

## 3. Results

Of the 30 refugee worker participants recruited, the majority of respondents were male and born in South Sudan (Table 2).

Vietnamese, Chinese, Arabic-speaking, and refugee workers had significantly lower mean job complexity, control, security, and overall job quality scale scores than workers born in Australia, New Zealand, India, and the Philippines (Table 3). Refugee workers were more likely to report that they had experienced bullying in the workplace compared with Caucasian workers and other ethnicity migrant workers, although the confidence intervals overlapped. With the exception of migrant workers from New Zealand, all refugee workers and all other migrant worker groups were statistically significantly more likely to report experiencing workplace racial discrimination in the last year compared with Caucasian workers born in Australia or workers born in New Zealand. Refugee workers were three times more likely to report having three or more workplace psychosocial stressors, although the confidence limits overlapped with other migrant groups.

### 3.1. Content Analysis

Content analysis identified six key content areas: underemployment, precarious work, financial pressure, unfair treatment, health and safety, and positive aspects of work. Table 4 shows the percentage of participants that made comments relating to each of the categories. Our subsequent analysis features interview extracts from a range of participants. To respect participant anonymity, extracts are not identifiable.

#### 3.1.1. Underutilised

Participants frequently spoke about the challenges they faced in securing employment and the ways in which they were not working at their full capacity. All participants were currently employed, but only 37% were in full-time employment (Table 2) and 43% of participants reported wanting to work more hours (Table 4). Forty percent of participants had tertiary qualifications (Table 2), although none of the participants who had migrated to Australia with a university degree had been able to acquire work related to their qualifications. Lack of degree equivalence and lack of Australian experience were commonly mentioned as reasons for this. As a consequence, participants spoke of taking less skilled jobs, with 57% of participants currently possessing educational qualifications that they were not using in their current jobs. One participant reported being advised to discount their qualifications as a way of gaining entry to the workforce: “Some people just advise me, when you apply, lower yourself, maybe you can get a job, then after that, when they know you, you can get a better job”. Decisions to take work that was not commensurate with skill level often resulted in feelings of disappointment: “… sometimes when I feel I have a qualification and I’m doing this job—I sometimes feel in myself unhappy—I should be doing a better job than this”, and the experience of stress: “… that’s where the stress comes from, you’ve got something to offer—and you are not allowed to”.

Thirty-seven percent of participants reported experiencing long-term unemployment (over 1 year) upon arrival to Australia (Table 4): “It’s very hard for people who come from different country and everything is new, and even they don’t know how to apply for a job”. Half of the participants reported taking on some type of unpaid work during periods of unemployment. For some, volunteer work was offered as a precursor to paid employment: “I was doing a trial for two weeks and I worked from 4 a.m. to 4 p.m. cleaning”. For others, volunteering was personally motivated and presented an opportunity to learn about the Australian work environment and a way of making a social contribution “I want to work there to get involved with the community more, to employ my language, and help people. It give good feeling, when your mood down or something, I feel better when you do such kind of work”.

#### 3.1.2. Precarious Work

Participants spoke about the tenuous nature of their working lives, and they did so in relation to their official employment status and being made to feel as though they could be easily replaced. Half of the participants made comments relating to their lack of job security (Table 4). As almost two-thirds of all participants were currently employed on a casual, part-time basis, or were self-employed, lack of security was a concern: “There is no security, I’ve been there for nearly one and a half years now, and I’m just casual, and the moneys very small and all that—how do you say that is a secure job?” For some, there was the additional fear of being fired: “Sometimes they just treat us—if you don’t work properly, like go home.” Participants speaking about lack of job security articulated worry and uncertainty about their futures. For some, these worries resulted in a willingness to accept a lower standard of treatment at work: “It was really tough, it was really hard … but I just kept quiet because fear of, if I lose this, I’m not going to get another one.”

Unstable work often coincided with experiences of questionable remuneration practices with 37% of participants speaking about issues of fair pay and 23% revealing that at one time or another, they had been victims of underpayment (Table 3). While unfair pay was a concern for many, for one young unskilled worker, there was a degree of resignation for this circumstance: “I feel like my job, I don’t have any qualification, I don’t have a degree, I don’t have nothing, It’s hard for me to demand, I need this, I need that—I feel like what I’m getting paid is not too bad.” When reflecting on their experiences of underpayment, examples often related to cash paying manual jobs (typically farm work/produce picking). Participants spoke about being paid amounts that were frequently less than half the current minimum wage and of being paid less than they had been promised. Participants often described feeling little recourse in these situations; for many, their response was to not return: “I just did it for two days and I was like, ‘Oh, if this is the only job in Australia I better stay home—or even go back to Africa’”.

#### 3.1.3. Financial Pressure

The combination of underemployment and precarious work created financial stress for many, with almost 40% of participants admitting that they did not earn enough to live on (Table 4): “When I came here sometimes I’ve got nothing to eat in the fridge … Oh god, how can I survive in this place”; 80% reported not earning enough for unexpected expenses: “I don’t have extra—I just live on the money that I earn”. Despite struggling themselves, two-thirds of participants sent remittances to family members in their home country “Sometimes when it comes to that [remittances] some days you go without—but you gotta help”. For many, this was a responsibility that accompanied their resettlement: “You got family back home, when they hear you’re in another country, the expectation is high—you need to help them”.

#### 3.1.4. Unfair Treatment

Participants frequently spoke about the ways in which they were treated differently or unfairly in the workplace, identifying supervisors, co-workers, and in some instances clients as responsible. Forty-three percent of participants spoke of experiencing bulling and 37% spoke of experiencing discrimination in the workplace (Table 4). Supervisors were sometimes identified as the source of unfair and differential treatment: “The supervisor was rude—if I look at it now, I feel like I’ve been bullied—the supervisor was really bad.” The inherent power ascribed to the supervisor led participants to respond in submissive or compliant ways: “The way they [the supervisor] spoke to the others was different to the way they spoke to me, so it kind of became a bit obvious, so I just kind of stayed out of the way”, “Some were saying you need to talk back to him, but I can’t—I have this fear that I can’t get another job—that’s why I just keep tolerating everything.” With respect to unfair treatment from co-workers, the level of action and response was less passive: “The cleaner left the bucket and mop with some dirty water—he [co-worker] took that dirty water and he poured on me, he poured everything on me—and I lost it”.

In some instances, clients and customers were identified as perpetrators of discrimination (Table 4). This was particularly the case for people working in health care and support roles: “There is some old people, they don’t want to be helped by someone black like me”, “Some of the residents I can clearly say they are racist—they don’t like the dark people around, and for us—easy they can accuse you of doing something, and once they report you, you are out of that section and that’s the warning for you”. While the working environment described thus far reflects adverse working conditions, there were also some positive and reassuring responses from participants.

#### 3.1.5. Health and Safety

Almost all participants reported having had adequate health and safety training in their current jobs (Table 3), and participants working in what we deem high-risk occupations (industrial cleaning, manufacturing, construction, electricity, gas, and waste services) reported being encouraged to use, and using, protective equipment. Participants often talked about the priority given to health and safety in their current jobs: “Our work they take safety—it’s like paramount”, “Training all the time, it’s really strict about the safety.” Some compared the Australian focus on health and safety with their home country: “It’s very strange for me that they worry so much about the back [safe lifting] because before we like to bend, when we cultivate—nothing happened to our mothers, to our grandmothers”, “When I start a new job, I see the induction which is good, I didn’t see this much in my county.” However, there were instances where participants had negative experiences in previous jobs: “The safety side with my first job was horrible—you can get killed over there.” Unlike the reports of unfair treatment mentioned above, participants reported moving on from jobs in unsafe work environments. 

#### 3.1.6. Positive Aspects

Participants reflected on the positive aspects of their jobs, namely the supportive relationships and the feelings of pride they took in their work. Positive relationships with both supervisors and co-workers were recounted: “The production manager, we joke all the time, it’s a really friendly environment”; “We always help each other. If anyone is struggling, you just help them.” Participants also found rewards in the work itself, particularly those working in social support roles: “I feel happy supporting people with disability—and, you know, help their needs—not because of the money but because I have the passion to help the people.” In many instances, positive aspects of work coexisted with negative aspects, with supportive co-workers making up for demanding supervisors: “Some of them like, if you feel a little bit stress, they encourage you, they talk to you and make you feel better”.

For some participants, periods of unemployment and lack of skill recognition lead to new careers. For example, eight participants retrained for jobs in the health care and social assistance sectors. For some, social assistance work was presented as an opportunity with government-assisted training programs. Despite the aforementioned experiences with discrimination and racism from clients, for many, working in a support role was a source of pride: “If someone waiting for you, and you have to shower them, or feed them, I feel I’m important, and I’m proud of what I’m doing”.

## 4. Discussion

The refugee sample was largely male, and just under half of the participants (48%) had a tertiary education. Almost two-thirds had lived in Australia for more than ten years, and many had had a range of work experiences. Comparison of workplace psychosocial stressors between refugee workers, Australian-born of Caucasian ancestry, and other migrant workers found Vietnamese, Chinese, and Arabic-speaking workers all had significantly lower mean scores for job complexity, control, security, and overall job quality, indicating that while they considered their jobs less complex, they felt they had less control in the work environment, that their jobs were less secure, and this resulted in an overall lower job quality score. With the exception of Australian-born of Caucasian ancestry and New Zealand workers, all other migrant workers were significantly more likely to experience racial discrimination. Refugee workers were three times more likely to have three or more workplace psychosocial stressors, although the confidence intervals overlapped with all the other worker groups. Given the small sample size of the refugee workers and the relatively small numbers who reported three or more psychosocial stressors, the large confidence intervals around all estimates made it difficult to identify any significant differences. Overall, the quantitative results provide some evidence that refugee workers experience the workplace differently than either Australian-born workers of Caucasian ancestry or other migrant workers.

Turning now to the content analysis of the refugee workers interviews, the responses here offer additional insights into the working conditions of this group. Participants reported being overqualified for the jobs they were currently employed in and reported a desire to work more hours, which is an indication that many were operating below their full working capacity. Underemployment in terms of hours worked and the under-use of skills have both been associated with poorer mental health outcomes for migrant workers [22,40]. Participants also reflected on insecurity over employment status, fear of being fired, and concern over fair pay. Some reported having been victims of wage theft [11]. Previous research in Australia identified wage theft as a significant problem for temporary migrant workers, but to date, this had not been identified as a particular problem for refugee workers. Many refugee workers not only need to support themselves; they need to provide support to family left behind. Refugee workers contribute both to the economy of the country they migrate to and to the economy of their country of birth [11].

One-third of participants were employed in the health care and social assistance industry, with most participants receiving government-assisted retraining in this field. While we refer to retraining as a positive aspect of the working experience, due to the employment outcome, we are mindful that refugees have historically been treated as Australia’s low skilled work force and that our refugee sample tends to support this trend of being relegated to low status and low paying jobs [18]. While a drop in occupation status has been found to accompany migration and subsequent occupation change, for refugee workers, this drop is far steeper in comparison to economic migrants [41]. Despite being directed into particular employment sectors, some participants found working in support roles to be inherently rewarding, which is a finding that would be interesting to explore in future research.

With regard to the experience of bullying and discrimination, we have reported higher rates of discrimination for all migrant workers compared with Australian-born workers of Caucasian ancestry. Granted that when asking about experience of discrimination, this question varied across the different surveys. Nonetheless, it is difficult to ignore the high rates of discrimination and bullying reported from our sample and the percentage of participants that spoke to these experiences. Approximately half of the South Sudanese refugee workers in this study reported experiencing bullying and/or discrimination in the workplace. Our previous research had already pointed to racialised discrimination as a potential workplace hazard [42], and our findings here strengthen these concerns. In recent years, Australia has seen an increase in reports of racism directed toward people from Africa, people who because of their “visible difference” may be easily identified as migrants and also presumed to be refugees. Given that negative attitudes toward refugees feature in mainstream Australian discourse [43], racialised discrimination based on migration status may be socially sanctioned. While treatment of the visibly different migrant has received some attention [44], a decade has now passed, and given our recent findings, it is perhaps time to revisit this topic.

Of relevance to our findings are the social theories of brain waste and social capital. Brain waste refers to the underutilisation of immigrant skills in the host country and specifically that high-level technical or professional skills are being wasted [45]. This was shown in the current study by reporting of being overqualified for their job and their relegation to low status and low-paying jobs. Brain waste can have adverse effects on both the individual (e.g., poorer mental health [22] by working in a job for which they are over skilled) and the economy of the host country through the undermining of positive self-selection of skilled migrants [46]. Lack of mainstream social capital was demonstrated in the current study through reports of unfair pay and treatment, wage theft, and job insecurity. However, the evidence related to immigrant or ethnic group social capital and labour market participation is mixed [47]. Some studies have found that immigrants with greater social capital have greater access to formal sector employment and higher earnings, while others argue that embeddedness in an ethnic social network precludes inclusion into the host society economy, while yet others argue about a “dark side” of exploitation rather than facilitation [48].

A review of recruiting research participants from refugee backgrounds recognised such research as having particular practical and ethical challenges [49]. Participants can be difficult to identify and engage in the research process [50]. Language and cultural differences can create barriers to engagement, as can participant vulnerability and confidentiality, particularly with respect to new arrivals leading to a difficulty in reaching and recruiting these groups. This study presented an additional level of complexity, despite not requiring workplaces to be being identified; our topic of inquiry would potentially illicit negative comments relating to work opportunities with former and/or current employers. Given the controversial ways in which humanitarian migrants are depicted and treated [51], reluctance to engage in research can be understood. The recruitment of a community leader assisted in reaching and engaging participants, but the small number attained over the time period available meant that only a rudimentary quantitative comparison could be conducted.

By conducting interviews in English, we were unable to recruit participants who were not proficient in English, which are participants who indeed may be the most vulnerable to negative workplace experiences. However, by accessing English-speaking participants, we were able to access people from a range of industries and able to discount the possibility that lack of language proficiency is a mediating factor in exposure to workplace stressors. The topic guide developed for this study focussed on aspects of the workplace, and while we were interested in exposure to psychosocial stressors, we did not attempt to assess mental health as has been done in previous research with other migrant groups [22,25]. The connection between negative experiences in relation to work and mental health has mixed findings. 

The strength of this study is that capturing both quantitative and qualitative data has allowed us to explore the refugee worker’s working experience in more detail and depth. Our qualitative analysis focussed on the key content areas from the interviews and identified important aspects of the refugee worker working experience that are not generally captured through the quantitative job quality measures. Although this analysis identified some possible indicators of poorer working conditions, it is important to note that we cannot make definitive claims about causation due to the scope of content analysis.

## 5. Conclusions

The findings from this sample indicate that when refugee workers find employment, they may find themselves working in challenging environments and exposed to higher levels of unfair treatment and psychosocial stressors compared with other migrant groups. This indicates that improvements are needed in the ways in which refugee workers are integrated into the workforce. People who migrate on humanitarian grounds need support not only in finding work, they need to be informed of their work rights and entitlements, as well as what constitutes fair work, so that they are able to recognise and potentially avoid unfair workplace practices. While proving information is important, this does not discount employer accountability. Employers who take on refugee workers have a responsibility to create a fair work environment that supports rather than hinders integration and wellbeing.

## Figures and Tables

**Table 1 ijerph-18-04023-t001:** Questions asked to determine workplace exposure to psychosocial hazards of working refugee participants (and used earlier in the cross-sectional surveys).

Psychosocial Hazard	Questions Asked
Job complexity	My job is more stressful than I ever imagined My job is complex and difficult
	My job requires learning new skills I use my skills in my current job
Control over work	I have freedom to decide how I do work I have a lot of say about what happens I have freedom to decide when I do work
Job security	I have a secure future in my job The company I work for will be in business in five years I worry about the future of my job
Salary	I get paid fairly for the things I do in my job
Bullying	Have you ever been bullied in the workplace?
Discrimination	Have you ever been treated unfairly by your employers or supervisors because of your country of birth? Have you ever been treated unfairly by your co-workers and colleagues because of your country of birth?
Open-ended questions	Has there been a time in your job when you were concerned about your safety but were afraid to voice your concern? In your job, do you receive support and encouragement from your co-workers?

**Table 2 ijerph-18-04023-t002:** Participants’ socio-demographic and employment characteristics by country of birth.

Demographic	Total	South Sudan n (%)	Afghanistan n (%)	Iraq n (%)
Total participants	30	20	8	2
Male	19 (63%)	14 (70%)	6 (75%)	0
Female	11 (37%)	7 (30%)	2 (25%)	2 (100%)
Age range (years)				
18–25	5 (17%)	2 (10%)	3 (37%)	0
26–35	10 (33%)	5 (25%)	4 (50%)	1 (50%)
36–45	9 (30%)	8 (40%)	1 (13%)	0
46–55	6 (20%)	5 (25%)	0	1 (50%)
Duration of residence in Australia				
0–5 years	6 (20%)	0	5 (62%)	1 (50%)
5–10 years	5 (17%)	2 (10%)	2 (25%)	1 (50%)
10+ years	19 (63%)	18 (90%)	1 (13%)	0
Highest educational attainment				
High school	5 (17%)	5 (25%)	0	0
Certificate/diploma	11 (37%)	7 (30%)	5 (62%)	0
Trade/apprenticeship	1 (3%)	1 (5%)	0	0
Bachelor degree or higher	13 (43%)	8 (40%)	3 (38%)	2 (100%)
Employment status				
Casual	9 (30%)	2 (6%)	5 (62%)	2 (100%)
Part-time	6 (20%)	5 (17%)	1 (13%)	0
Full-time	11 (37%)	10 (33%)	1 (13%)	0
Self Employed	4 (13%)	3 (10%)	1 (13%)	0
Industry of employer				
Construction/Trade	2 (6%)	1 (3%)	1 (13%)	0
Food services	9 (30%)	1 (3%)	6 (75%)	2 (100%)
Education and training	2 (7%)	2 (6%)	0	0
Support services	1 (3%)	0	1 (13%)	0
Health care and social assistance	10 (33%)	10 (33%)	0	0
Cleaning	2 (7%)	2 (6%)	0	0
Warehousing	2 (7%)	2 (6%)	0	0
Other	2 (7%)	2 (6%)	0	0
Size of employer				
5–19 employees	10 (38%)	4 (15%)	4 (51%)	2 (100%)
20–199 employees	13 (50%)	11 (42%)	2 (26%)	0
200 or more employees	3 (12%)	2 (8%)	1 (13%)	0

**Table 3 ijerph-18-04023-t003:** Estimates (expressed as means or percentages with 95% CI) for workplace psychosocial stressors for Caucasian Australian ^a^ and migrants workers from six countries ^a^ and refugee workers ^b^.

Workplace Psychosocial Stressors	Caucasian Australia	New Zealand	India	Philippines	Vietnam	China	Arabic Speaking	Refugee Workers
Complexity ^1^ scale Mean [95% CI]	16.9 [16.7,17.1]	17.5 [17.2,17.8]	17.4 [17.2,17.7]	17.5 [17.1,17.8]	11.8 [11.1,12.4]	10.9 [10.3,11.4]	11.3 [10.7,11.8]	12.32 [10.5,14.1]
Control ^1^ scale Mean [95% CI]	13.7 [13.5,14]	14.0 [13.6,14.3]	13.9 [13.6,14.3]	13.9 [13.5,14.3]	12.9 [12.3,13.4]	9.2 [8.5,9.8]	10.8 [10.2,11.4]	10.5 [6.9,14.1]
Security ^1^ scale Mean [95% CI]	16.3 [16,16.5]	16.1 [15.8,16.4]	15.2 [14.9,15.5]	16.0 [15.7,16.4]	12.0 [11.5,12.5]	10.8 [10.5,11.2]	11.5 [11,11.9]	9.5 [7.6,11.3]
Job quality ^1^ Mean [95% CI]	52.4 [51.6,53.1]	53.2 [52.2,54.2]	52.2 [51.3,53.1]	52.8 [51.9,53.7]	38.5 [35.8,41.2]	38.3 [36.2,40.4]	36.0 [33.9,38.0]	35.8 [30.2,41.4]
Unfair pay % [95% CI]	35.3 [31.3,39.7]	29.3 [24.4,34.6]	30.6 [25.7,36.0]	24.5 [20.1,29.5]	22.3 [15.1,31.6]	29.0 [19.0,41.6]	26.8 [19.2,36.2]	35.5 [28.5, 42.5]
Bullied ^2^ % [95% CI]	9.5 [7.1,12.6]	11.5 [8.5,15.4]	11.5 [8.0,16.4]	10.5 [7.7,14.2]	13.7 [6.6,26.3]	16.3 [8.8,28.0]	10.2 [5.8,17.3]	20.1 [4.8, 35.2]
Discriminated in last year % [95% CI]	1.9 [0.9,3.7]	8.0 [5.2,12.2]	20.7 [16.1,26.4]	15.7 [12.2,20.0]	25.3 [16.5,36.8]	24.8 [16.5,35.5]	31.5 [22.5,42.1]	23.3 [7.3,39.4]
Three or more psychosocial stressors % [95% CI]	10.9 [8.5,14.1]	12.5 [9.0,17.1]	13.6 [10.0,18.3]	11.0 [8.0,14.8]	9.5 [4.2,20.2]	13.7 [6.9,25.4]	8.1 [4.6,13.8]	30.0 [12.6, 47.4]

^a^ Migrants of Vietnamese, Chinese, or Arabic speaking ancestry and workers born in India, New Zealand, and the Philippines ^b^ Migrants who entered Australia on humanitarian grounds. ^1^ Higher scores reflect higher job complexity, control, security, and overall job quality. Job quality is the sum of the scores from complexity (range 0–24), control (range 0–18), security (range 0–18), and unfair pay (range 0 = 6) for a total job quality score with a range of 0–66. ^2^ Humanitarian migrants were asked if they had ever been bullied, whereas the other migrant groups were asked within the last 6 months.

**Table 4 ijerph-18-04023-t004:** Content analysis summary (expressed as a percentage of total participants and country of birth) reporting on each topic.

Content Category		Total (*n* = 30) %
Underutilised	Desire to work more hours	43
	Overqualified for current job	57
	Volunteer work	50
	Long-term unemployment (over 1 year)	37
Precarious work	Lack of job security	50
	Fear of being fired	27
	Unfair pay	37
	Underpayment	23
Financial pressure	Not earning enough to live on	37
	Not earning enough for unexpected expenses	80
	Sending remittances home	67
Unfair treatment	Bullying	43
	Discrimination	37
	Negative supervisor relationships	13
	Negative co-worker relationships	23
	Negative client/customer relationships	17
	Feeling defenceless at work	30
Health and safety at work	Adequate health and safety training	93
	Use protective equipment	67
	Health and safety at risk	13
Positive work aspects	Positive supervisor relationships	63
	Positive co-worker relationships	63
	Positive client/customer relationships	10
	Feeling (self) pride at work	87
	Feeling respected at work	83
	Learning new skills (retraining)	57

## Data Availability

The data supporting the finding of this study are available on request from the corresponding author, [A.R.]. The data are not publicly available due to their containing information that could compromise the privacy of research participants.

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
