# Peer review of "How Refugees Experience the Australian Workplace: A Comparative Mixed Methods Study"

_ijerph, 2021, doi:10.3390/ijerph18084023_

Round 1

Reviewer 1 Report

I would like to thank the authors for their efforts to zoom into the subject matter of the employment of refugees in Australia. The title of the study captures the thematic focus. The relationship between the six migrant groups (n=1062) and humanitarian migrants (n=30) is not clear. Subsequent text suggests that 30 refugees represent the  unit of analysis. Divided across three countries of origin (do they correspond to the top countries of origin of all refugees globally or in Australia?), the actual sample per country is tiny (for example,  2 Iraqi). This suggests skewed results in terms of the refugees’ profile. It is not clear whether quantitative analysis adds any value to the research. Perhaps focus more on the qualitative analysis of responses?

A structured abstract offers an overview of the background, objectives, methods, results, conclusions, and recommendation.

Introduction: offers a balanced overview and analysis of the literature, with specific links to the relevance thereof for the study. 

26-27: a suggestion to use more recent data on the number of international migrants globally. In addition to 2017 estimates, 2019 estimates are available, with 272 million. For details, please see the International Migrant Stock 2019, a dataset released by the Population Division of the UN Department of Economic and Social Affairs (DESA).

31: the link to the UNHCR source gives 79,5 million forcibly displaced persons with a higher number for each category included. this is not a problem as the reference includes a data of access. May be still to include a reference to a stable source? Or update the numbers per source?

36: these visa holders refers to?

37: to specify top sending countries to Australia?

  1. Materials and methods

2.1. Participants recruitment describes the participants group and its features.

Data collection details offers an excellent and detailed overview of the interview questions.

Definitions and the reasoning behind the order of questions represent a welcome addition. 

 2.2. Analysis section explains the coding.

3 Results

190: is it the first time that the size of the sample is mentioned (after the abstract)?

4 Discussion assesses the results of the interviews.

5 Conclusions contain recommendations for the employers. The grounds for the conclusion that refugees find themselves working in challenging environments and exposed to higher levels of unfair treatment and psychosocial stressors compared with other migrant groups are not clear.

Author Response

Response to reviewers

REVIEWER ONE

I would like to thank the authors for their efforts to zoom into the subject matter of the employment of refugees in Australia. The title of the study captures the thematic focus. The relationship between the six migrant groups (n=1062) and humanitarian migrants (n=30) is not clear. 

Response: Thanks for your comments. The aim of the paper was to compare the working conditions of Refugee workers in Australia with those of other migrant workers who did not arrive as refugees, and a group of Australian-born workers of Caucasian ancestry.  Historically, Australia has relegated refugees to work in poorer quality jobs and we wanted to examine this in relation to the working conditions of other migrant worker groups and Australian-born workers.

Subsequent text suggests that 30 refugees represent the  unit of analysis. Divided across three countries of origin (do they correspond to the top countries of origin of all refugees globally or in Australia?), the actual sample per country is tiny (for example,  2 Iraqi). This suggests skewed results in terms of the refugees’ profile.

Response: Refugees from Iraq and Afghanistan are represented in the top five countries of origin in the past five years. Refugees from South Sudan were frequent arrivals in Australia during the years  2000 to 2016. We agree with the reviewer’s comments about skewed results in terms of the refugees profile, because of small numbers. We have amended Table 4 to show only the summary results – not those for each group separately.

It is not clear whether quantitative analysis adds any value to the research. Perhaps focus more on the qualitative analysis of responses?

Response: As I mentioned above, the aim of this paper was to compare refugee working conditions with those of Australian born and migrant workers from other countries. We feel it is important to compare these groups in order to highlight the disparities (if any) between them.

A structured abstract offers an overview of the background, objectives, methods, results, conclusions, and recommendation.

Introduction: offers a balanced overview and analysis of the literature, with specific links to the relevance thereof for the study. 

26-27: a suggestion to use more recent data on the number of international migrants globally. In addition to 2017 estimates, 2019 estimates are available, with 272 million. For details, please see the International Migrant Stock 2019, a dataset released by the Population Division of the UN Department of Economic and Social Affairs (DESA).

Response: Thank you for pointing this out – we have updated these numbers (and the citation) in the introduction (lines 26-27).

31: the link to the UNHCR source gives 79,5 million forcibly displaced persons with a higher number for each category included. this is not a problem as the reference includes a data of access. May be still to include a reference to a stable source? Or update the numbers per source?

Response: Thanks, we have updated those numbers too.

36: these visa holders refers to?

Response: this phrase has been removed

37: to specify top sending countries to Australia?

Response: The first paragraph of the introduction has been amended as below to reflect all of the above changes.

 Refugees and asylum seekers make up an increasing component (29%) of the global migrant population of 272 million {United Nations, 2019 #88}. The United Nations High Commissioner for Refugees (UNHCR) estimates that there are 79.5 million forcibly displaced persons, consisting of 45.7 million internally displaced people, 26 million refugees (those who have had their claim for refugee status recognised under the 1951 Convention) and 4.2 million asylum seekers (those who have not yet had their claim for refugee status recognised) {UNHCR, 2019 #2}. Sixty-eight percent of refugees currently come from five countries, Syria, Venezuela, Afghanistan, South Sudan and Myanmar{UNHCR, 2019 #2}. Australia currently settles over 12,000 refugees annually as part of its humanitarian program, and has done so, in varying numbers since the first 170,000 people were accepted as part of the displaced persons program established at the end of World War II {Phillips, 2015 #3}. In the 2018-19 period, Iraq, the Democratic Republic of Congo, Myanmar, Syria and Afghanistan were the top five sending countries.{SSI, 2019 #4}. Most people arrived in Australian from South Sudan during the period 2001-2006{Refugee Council of Australia, 2018 #89}.

"Materials and methods
2.1. Participants recruitment describes the participants group and its features.

Response: No response required

Data collection details offers an excellent and detailed overview of the interview questions.

Response: No response required

Definitions and the reasoning behind the order of questions represent a welcome addition.

Response: No response required

2.2. Analysis section explains the coding.

Response: No response required

3 Results
190: is it the first time that the size of the sample is mentioned (after the abstract)?

Response: Yes, but this is normal practice – to state the number of recruited participants in the results section – rather than anywhere else in the document.

4 Discussion assesses the results of the interviews.

Response: No response required

5 Conclusions contain recommendations for the employers. The grounds for the conclusion that refugees find themselves working in challenging environments and exposed to higher levels of unfair treatment and psychosocial stressors compared with other migrant groups are not clear."

Response: We showed in Table 3 (new Table 3) that the prevalence of experiencing 3 or more psychosocial stressors at work was higher among refugee workers (30%), than any other group.  Similalry, their exposure to bullying and discrimination (unfair treatment) was higher than in other groups (also shown in Table 3).

Reviewer 2 Report

Overall, this is an interesting study that I believe contributes to the field.  My primary concerns are related to the descriptions of the three groups which were compared, and the need to further explain why these particular groups were established in both the introduction and methods. 

Review

The biggest need for improvement revolves around explaining the three groups that were compared.   explain further why these different groups were established: refugees, workers of primarily Asian and Arabic ancestry, and Australian-born Caucasian workers.  Why were the groups broken down in this way?  Are there significant differences in wealth, educational attainment, or employment opportunities?  Among the second group with immigrants of Vietnamese et al ancestry, were these workers immigrants themselves or Australian-born? 

Why were Vietnamese, Chinese or Arabic speaking ancestry compared to workers botn in India or New Zealand or the Philippines?  Does ancestry imply that these participants were born in Vietnam, China, etc?  It seems that comparing workers born in specific countries to those with ancestry (specifically "Arabic speaking ancestry") is not a direct comparison and may be inappropriate.  More information is needed to describe these groups.

Define humanitarian workers in the introduction- I think of this as emergency aid or relief workers, not as migrant workers.  Clarification of terms would be helpful.

It would also be helpful to clarify the following: "Australian-born workers of Caucasian ancestry" - implies that there are NO non-Caucasian Australian-born workers, which is not the case.  Was race used as demographic?  If so, please take care to ensure that race is not used interchangeably with immigrant.

Page 2 lines 45-46 - elaborate on the principle that some temporary workers (such as the backpackers and international students mentioned) will differ from refugee workers in important ways

Page 2 - as above, explain further why these different groups were established: refugees, workers of primarily Asian and Arabic ancestry, and Australian-born Caucasian workers.  Why were the groups broken down in this way?  Are there significant differences in wealth, educational attainment, or employment opportunities?  Among the second group with immigrants of Vietnamese et al ancestry, were these workers immigrants themselves or Australian-born? please elaborate on why these decisions were made and chosen for different groups here.

Participant recruitment and data collection - consider additional subheadings in this section.  This section could also be more concise, particularly the paragraphs about bullying.  Consider a table with questions rather than listing the questions in the text of this section.  Please clarify why quantitative and qualitative surveys defined bullying differently.

In section 2.2, analysis, please define terms to increase the applicability and user friendliness of the article - Iterative Proportional Fitting, Triangulation Design: Convergence Model.  They do not need to be long definitions, but a sentence summarizing each at minimum. 

Results

Consider creating a table identifying themes with select quotations.

Author Response

REVIEWER TWO

Overall, this is an interesting study that I believe contributes to the field.  My primary concerns are related to the descriptions of the three groups which were compared, and the need to further explain why these particular groups were established in both the introduction and methods. 

Review

The biggest need for improvement revolves around explaining the three groups that were compared.   explain further why these different groups were established: refugees, workers of primarily Asian and Arabic ancestry, and Australian-born Caucasian workers.  Why were the groups broken down in this way?  Are there significant differences in wealth, educational attainment, or employment opportunities?  Among the second group with immigrants of Vietnamese et al ancestry, were these workers immigrants themselves or Australian-born? 

Response: There is no sampling frame of migrants in Australia. In order to obtain our sample, we identified the 20 most common surnames for each group, as well as area of residence, and linked that to the electronic white pages to call households of our target groups. We first attempted this method among workers of Arabic speaking, Vietnamese and Chinese ancestry. We were unsure how many participants we would reach, so we widened our target population to include second generation migrants and greater, instead of limiting it to those born in Vietnam, China or a list of countries where Arabic was spoken (e.g. Egypt, Syria, Saudi Arabia, Sudan etc). However, an analysis of our data showed that most of our participants were in fact first generation migrants, with 29% of the Arabic-speaking, 23% of the Vietnamese and 14% of the Chinese workers being born in Australia. So, we used this same recruitment method for our next survey examining exposure to workplace hazards among workers born in India, the Philippines and New Zealand, and this time only recruiting participants currently working in Australia but who were born in India, the Philippines or New Zealand.

Why were Vietnamese, Chinese or Arabic speaking ancestry compared to workers botn in India or New Zealand or the Philippines?  Does ancestry imply that these participants were born in Vietnam, China, etc?  It seems that comparing workers born in specific countries to those with ancestry (specifically "Arabic speaking ancestry") is not a direct comparison and may be inappropriate.  More information is needed to describe these groups.

Response: These groups of workers (from China, Vietnam, the Philippines. India, New Zealand) were examined in our earlier studies for several reasons. Earlier work of ours had highlighted that workers from New Zealand had higher work-related injuries and fatalities than Australian-born workers, so we wanted to explore their working conditions in more detail. New Zealanders enter Australia on a different visa type, than the skilled workers coming from China, Vietnam, the Philippines and India. Additionally, these groups of workers come from among the top 10 countries Australia receives migrants from. We included Arabic speaking workers because our earlier focus group discussions highlighted that these workers were particularly disadvantaged in the workplace and we wanted to explore that in more detail in a quantitative survey examining their exposure to hazards at work.

Define humanitarian workers in the introduction- I think of this as emergency aid or relief workers, not as migrant workers.  Clarification of terms would be helpful.

 Response: We have removed the term humanitarian worker and replaced it with refugees – the abstract has been amended as set out below.

There is a growing body of evidence indicating poorer working conditions for migrant workers, particularly refugees, compared with native-born workers. Our objectives were to compare exposure to workplace psychosocial stressors in working refugees with other migrant groups and Australian-born workers of Caucasian ancestry and to describe the working experience of refugees. Cross-sectional surveys collected information on the workplace stressors of job complexity, control, security, bullying and racial discrimination from six migrant groups (n=1062) and Caucasian Australians (n=1051); semi-structured face-to-face interviews were used with currently employed refugees (n=30).  Content analysis examined the qualitative data. Compared to all other groups, working refugees were more likely to report experiencing racial discrimination in the workplace and to report exposure to more than three hazards. Content analysis indicated that working refugees are working below their capacity, in terms of hours and qualifications, and in jobs that were low status and lacked security. Despite challenging work conditions, participants reported adequate health and safety training and feeling a sense of pride in their work. These findings highlight the need for better support for refugees in negotiating the workplace once they find employment, and the importance of employers providing an inclusive and equitable workplace.

It would also be helpful to clarify the following: "Australian-born workers of Caucasian ancestry" - implies that there are NO non-Caucasian Australian-born workers, which is not the case.  Was race used as demographic?  If so, please take care to ensure that race is not used interchangeably with immigrant.

 Response: We included an Australian-born group of Caucasian ancestry because in our earlier focus group discussions and in-depth interviews, second-generation migrants, of non-Caucasian ancestry, reported barriers in the workplace that they perceived did not occur among those with Caucasian ancestry.

We are definitely not implying that there are no non-Caucasian Australians.

Page 2 lines 45-46 - elaborate on the principle that some temporary workers (such as the backpackers and international students mentioned) will differ from refugee workers in important ways

 Response: Lines 58-66 have been amended as shown below.

There are a range of reasons why refugees might be more likely to incur poorer working conditions compared with other groups of migrants (e.g. skilled migrants, temporary migrants or international students) or native-born workers. These include: segregation into niche occupations and industries {Colic‐Peisker, 2007 #58;Teicher, 2002 #23} and non-recognition of prior educational qualifications {Colic-Peisker, 2007 #22;Jupp, 2002 #24}, lack of mainstream social networks to assist in job searching {Colic‐Peisker, 2007 #58}, poorer English language proficiency as well as the sending home of remittances necessitates risk taking and working in survival jobs {Reid, 2014 #6}.

Page 2 - as above, explain further why these different groups were established: refugees, workers of primarily Asian and Arabic ancestry, and Australian-born Caucasian workers.  Why were the groups broken down in this way?  Are there significant differences in wealth, educational attainment, or employment opportunities?  Among the second group with immigrants of Vietnamese et al ancestry, were these workers immigrants themselves or Australian-born? please elaborate on why these decisions were made and chosen for different groups here.

 Response: The following paragraph has been added between lines 101 and 115 to clarify why these groups have been chosen to compare working conditions with working refugees.

These groups of workers were chosen as suitable comparisons to the working refugee group for several reasons. 1) India, China, Vietnam, the Philippines and New Zealand are in the top 10 group of countries Australia receives migrants from. 2) Earlier focus group discussions we had undertaken highlighted that workers of Arabic-speaking ancestry were particularly disadvantaged in the workplace, in terms of exposure to hazards, and this was confirmed in our quantitative survey{Boyle, 2015 #26;Boyle, 2015 #52}. 3) Similarly, earlier work of ours had shown that workers born in New Zealand had higher hospitalisations for work-related injuries and higher work-related fatalities than Australian-born workers or workers born in any other country{Reid, 2016 #25}. New Zealanders enter Australia on a different visa type, than the skilled workers coming from China, Vietnam, the Philippines and India. 4) We included an Australian-born group of Caucasian ancestry because in our earlier focus group discussions and in-depth interviews, second-generation migrants, of non-Caucasian ancestry, reported barriers in the workplace that they perceived did not occur among those with Caucasian ancestry{Reid, 2014 #6}.

Participant recruitment and data collection - consider additional subheadings in this section.  This section could also be more concise, particularly the paragraphs about bullying. 

Response: The sections on recruitment and data collection have been separated into two subsections. The section outlining the methods used to collect information on bullying has been shortened and clarified as outlined below.

Refugee participants were asked about bullying (Have you ever been bullied in the workplace?) and discrimination in the workplace (Have you ever been treated unfairly by your employers or supervisors because of your country of birth? Have you ever been treated unfairly by your co-workers and colleagues because of your country of birth?). Responses for perceived bullying and discrimination were coded as yes/no. These responses were compared with questions collected in the three cross-sectional surveys in which respondents were asked if they had ever been bullied within the last six months in the workplace and been discriminated against in the workplace due to race or ethnicity in the last year.

Consider a table with questions rather than listing the questions in the text of this section

Response:  Table 1 has been included to display the questions, rather than including them in the text.  The relevant text has been amended as necessary

Table 1. Questions asked to determine workplace exposure to psychosocial hazards of working refugee participants (and used earlier in the cross-sectional surveys).

Psychosocial hazard

Questions asked

Job complexity

My job is more stressful than I ever imagined

My job is complex and difficult

My job requires learning new skills

I use my skills in my current job

Control over work

I have freedom to decide how I do work

I have a lot of say about what happens

I have freedom to decide when I do work

Job security

I have a secure future in my job,

The company I work for will be in business in five years,

I worry about the future of my job

Salary

I get paid fairly for the things I do in my job

Bullying

Have you ever been bullied in the workplace

Discrimination

Have you ever been treated unfairly by your employers or supervisors because of your country of birth?

Have you ever been treated unfairly by your co-workers and colleagues because of your country of birth?

Open-ended questions

Has there been a time in your job when you were concerned about your safety but were afraid to voice your concern?

In your job do you receive support and encouragement from your co-workers?

In section 2.2, analysis, please define terms to increase the applicability and user friendliness of the article - Iterative Proportional Fitting, Triangulation Design: Convergence Model.  They do not need to be long definitions, but a sentence summarizing each at minimum. 

 Response: The first paragraph of the data analysis section has been amended to read the following;

The three cross-sectional surveys were weighted using a technique that weights the sample to the same proportions found in each migrant population (Iterative Proportional Fitting) {Kolenikov, 2014 #1053}. We weighted each migrant group for sex, age, education and area of residence produce population estimates of workplace psychosocial stressors. Due to sample size differences, percentages and means with 95% confidence intervals were produced for comparison with the current study results. The confidence intervals were used to indicate statistically significant differences. The mixed method approach used in this study was that of Triangulation Design: Convergence Model, which does not require the integration of the different data sets for analysis {Creswell, 2016 #87}. We used semi-structured interviews to collect qualitative data and then compared the results from the qualitative interviews with the results we already had from previous cross-sectional studies[38].

Results

Consider creating a table identifying themes with select quotations.

Response: We have not done this as we wanted to show the quotes in relationship to our discussion – rather than having them shown elsewhere in the paper and in table format.

Submission Date

22 February 2021

Date of this review

02 Mar 2021 19:44:15

Bottom of Form

© 1996-2021 MDPI (Basel, Switzerland) unless otherwise stated

Reviewer 3 Report

From 197 to 200 there is a free space

Table 3 all in a complete data are not seen.

They should improve the presentations of all references (474-598)

They should improve the presentation of table 4

They should put references in paragraph (340-357)

They must put the references well:

  • 379-379
  • 394
  • 396-397

In general they should check the references in the text of the manuscript. Since they start with reference number 2 and number 1 does not appear, nor does numbers: 3, 8, 35, 48, 54, 55.

It would be convenient for them to send the informed consent provided to the participants.

Author Response

From 197 to 200 there is a free space

Response: - Table is now in landscape format 

Table 3 all in a complete data are not seen.

Response: - Table is now in landscape format 

They should improve the presentations of all references (474-598)

They should improve the presentation of table 4

Response: Table 4 now centred in the middle of the page

They should put references in paragraph (340-357)

Response: There are no references in those paragraphs - we are presenting our results

They must put the references well:

  • 379-379

Response: Fixed - thanks

  • 394
  • 396-397

Response: Fixed - thanks

In general they should check the references in the text of the manuscript. Since they start with reference number 2 and number 1 does not appear, nor does numbers: 3, 8, 35, 48, 54, 55.

Response: References in text start with number 1. All the other reference numbers listed above are included in the text - except 54 and 55 (references only go up to number 51).

It would be convenient for them to send the informed consent provided to the participants.

Response: Informed consent form attached

Round 2

Reviewer 1 Report

The main concern is the size of the sample. The conclusions are generic and not linked to the study.

Author Response

Response to reviewer

Thank you for the opportunity to respond to the reviewer’s comments.  We have made changes in response to them as outlined below.

Reviewer: The main concern is the size of the sample.

Response: Sample size in qualitative studies has long been the subject of discussion. Unlike quantitative research, there is no sample size calculation that can be performed to determine the ‘correct’ number of participants who must be recruited.  We based our sample size of informational redundancy[1]. After conducting 30 interviews, we believed that no new information would be obtained by conducting any more interviews.  This was not an a priori decision, but was informed by the quality and content of the data that was being collected from participants.

The following has been added to lines 72 and 73 of the text.

Participant recruitment ceased when no new information was obtained by completing more interviews[1].

Reviewer: The conclusions are generic and not linked to the study.

Response:  We have restated the main findings of this study in the conclusion and then broadened our discussion to discuss ways to ameliorate the hazards experienced by refugee wokers in the workplace.

  1. Vasileiou, K.; Barnett, J.; Thorpe, S.; Young, T. Characterising and justifying sample size sufficiency in interview-based studies: systematic analysis of qualitative health research over a 15-year period. BMC Med Res Methodol 2018, 18, 148, doi:10.1186/s12874-018-0594-7.

Round 3

Reviewer 1 Report

I would like to thank the authors for the additional efforts and for addressing the comments and suggestions. The text reads better, and updated data represent a welcome addition. Conclusions are well balanced and address the accountability of the employers.